# The widespread and unjust drinking water and clean water crisis in the United States

J. Tom Mueller [1][✉] & Stephen Gasteyer[2]

Many households in the United States face issues of incomplete plumbing and poor water quality. Prior scholarship on this issue has focused on one dimension of water hardship at a time, leaving the full picture incomplete. Here we begin to complete this picture by documenting incomplete plumbing and poor drinking water quality for the entire United States, as well as poor wastewater quality for the 39 states and territories where data is reliable. In doing so, we find evidence of a regionally-clustered, socially unequal household water crisis. Using data from the American Community Survey and the Environmental Protection Agency, we show there are 489,836 households lacking complete plumbing, 1,165 community water systems in Safe Drinking Water Act Serious Violation, and 9,457 Clean Water Act permittees in Significant Noncompliance. Further, elevated levels of water hardship are associated with rurality, poverty, indigeneity, education, and age—representing a nationwide environmental injustice.

[1] Department of Sociology, Social Work, and Anthropology, Utah State University, Logan, UT, USA. [2] Department of Sociology, Michigan State University, East Lansing, MI, USA. [✉]email: Tom.Mueller@usu.edu

Both in and out of the country, most presume that residents of the United States live with close to universal access to potable water and sanitation. The United Nations Sustainable Development Goals Tracker, which tracks progress toward meeting Sustainable Development Goal Number 6—calling for universal access to potable water and sanitation for all by 2030—estimates that 99.2% of the US population has continuous access to potable water and 88.9% has access to sanitation[1]. By percentages and the lived experience of most Americans, this appears accurate. The American Community Survey shows that from 2014 to 2018 only an estimated 0.41% of occupied US households lacked access to complete plumbing—meaning access to hot and cold water, a sink with a faucet, and a bath or shower. Although this relative percentage may be low, this 0.41% corresponds to 489,836 households spread unevenly across the country, making the absolute number quite troubling. These numbers become even more dramatic when we broaden our scope to poor household water quality, where the estimates we provide in this paper show the issue affects a far greater share of the population (Table 1).

This study builds on a growing body of evidence showing access to plumbing, water quality, and basic sanitation are lacking for a disturbingly large number of US residents by providing a definitive picture of the ongoing household water crisis in the United States. Water and sanitation issues have been a growing concern in the United States, particularly among policy organizations, for the past 20 years[2–10]. For example, the now-dated Still Living without the Basics report used Census data from 2000 to show that more than 670,000 households (0.64% of households and 1.7 million people) lacked access to complete plumbing facilities[7]. Further, the Water Infrastructure Network published a report in 2004 citing a gap of $23 billion between available funding and needed water and sanitation infrastructure investments[6]. In line with this, the American Society of Civil Engineers has repeatedly given the United States a "D" grade for water infrastructure, and "D-" for wastewater infrastructure in their annual "Infrastructure Report Card"[11]. Although water hardship in the United States has experienced some academic attention, much of the work has become dated and has generally focused on a single dimension of the issue at a time—for example, recent scholarship has focused on exclusively incomplete plumbing[3,4,9], water quality[5,10], or on only urban parts of the country[2]. This has left our understanding of the scope of the issue incomplete. In this paper, we estimate and map the full scope of water hardship for the dimensions of incomplete plumbing and poor drinking water quality across the entire United States, while also estimating and mapping the scope of poor wastewater quality for the 39 states where EPA data is reliable, in order to complete this picture.

Prior work from academics and policy groups on dimensions of water hardship has found water access issues pattern along common social inequalities in the United States. The Natural Resources Defense Council released a report demonstrating the disproportionate impact on people of color posed by Safe Drinking Water and Clean Water Act regulatory burdens[12], which built on similar peer reviewed findings[13,14]. Furthermore, both policy papers and peer reviewed studies have analyzed Census data to estimate the population lacking access to complete plumbing facilities and clean water[2–10,12]. The studies suggest low-income and non-White people—particularly indigenous populations who continue to face injustices related to legacies of settler colonialism[15]—are significantly more likely to have incomplete plumbing and unclean water[3,12]. Further, it appears incomplete plumbing may be a disproportionately rural issue, while poor water quality may be a disproportionately urban issue[5,9]. Direct comparisons, as we perform here, are needed to fully establish the variability of this inequality between dimensions of water hardship.

The prior scholarship on the inequitable distribution of plumbing and pollution speaks to the well-documented environmental injustices found throughout the United States. Environmental injustice, meaning the absence of "fair treatment and meaningful involvement of all people regardless of race, color, national origin, or income with respect to the development, implementation, and enforcement of environmental laws, regulations, and policies" (p. 558)[16], has been documented in the United States along the social dimensions of income[17,18], poverty[19], race and ethnicity[20,21], age[22], education[22,23], and rurality[22,24,25]. Based on the evidence of prior work on water hardship, it is clear household water access represents an ongoing environmental injustice in the United States[5]. However, the specific dimensions of this injustice, and how they vary between type of water hardship remain largely unknown. To address this gap, we estimate models of water injustice for the previously identified social dimensions at the county level for elevated levels of both incomplete plumbing and poor water quality.

## Results

**Level of water hardship in the United States.** Based upon the most recent available data reported by both the United States Census Bureau via the American Community Survey and the Environmental Protection Agency via Enforcement and Compliance History Online, we find that incomplete plumbing and poor water quality affects millions of Americans as of 2014–2018 and August 2020, respectively (Table 1)[26,27]. A total of 0.41% of households, or 489,836 households, lacked complete plumbing from 2014–2018 in the United States. Further, 509 counties, representing over 13 million Americans, have an elevated level of the issue where >1% of household do not have complete indoor plumbing (Table 2). Thus, even if individuals are not experiencing the issue themselves, they may live in a community where incomplete plumbing is a serious issue.

The portion of the population affected by poor water quality is much greater than that of incomplete plumbing. Poor water quality in our analysis is indicated in two ways, (1) Safe Drinking Water Act Serious Violators and (2) Clean Water Act Significant Noncompliance. For the first, community water systems are regulated under the Safe Drinking Water Act and are scored based on their violation and compliance history, those community water systems that are the most problematic are recorded as Serious Violators by the Environmental Protection Agency[27]. Second, any facility that discharges directly into waters in the United States is issued a Clean Water Act permit. Those which "hold a more severe level of environmental threat" are ruled as being in Significant Noncompliance[27]. Importantly, although data on Safe Drinking Water Act Serious Violators is available nationwide, the Clean Water Act data reported by the EPA is known to be inaccurate for 13 states. Thus, although we can draw national conclusions for incomplete plumbing and Safe Drinking Water Act violations, our understanding of Clean Water Act violations is limited to the 39 states and territories for which data are available and reliable.

Using these two measures of poor water quality, we find 2.44% of community water systems, a total of 1165, were Safe Drinking Water Act Serious Violators and 3.37% of Clean Water Act permittees in the 39 states and territories with accurate data (see Methods for more details), a total of 9457, were in Significant Noncompliance as of 18 August 2020. At the county level, this corresponds to an average of 2.86% of county community water systems being listed as Safe Drinking Water Act Significant Violators and an average of 6.23% of county Clean Water Act permittees being listed as Significant Noncompliers. Due to limitations in the data, we are unable to determine exactly how many individuals are linked to each problematic community water system or Clean Water Act permittee, however, we do find that over 81 million

**Table 1 Estimates of water hardship in the United States.**

|  | Total | Mean | SD | Min | Max | N(counties) |
|---|---|---|---|---|---|---|
| Percent of households without complete plumbing | 0.41 | 0.66 | 1.44 | 0.00 | 35.41 | 3220 |
| Percent of community water systems listed as SDWA Significant Violator | 2.44 | 2.86 | 9.32 | 0.00 | 100.0 | 3144 |
| Percent of permittees listed as CWA Significant Noncomplier | 3.37 | 6.23 | 8.54 | 0.00 | 100.0 | 2262 |

Note: total number of counties varies due to some counties having no reporting utilities and the dropping of 13 states with Clean Water Act data issues.

**Table 2 Estimates of elevated levels of water hardship in the United States.**

| Counties with greater than one percent of... | N(counties) | Percent | Population |
|---|---|---|---|
| Households with incomplete plumbing | 509 | 15.81 | 13,103,341 |
| Community water systems listed as SDWA Significant Violators | 596 | 18.96 | 81,627,967 |
| CWA permittees listed as CWA Significant Noncomplier | 1455 | 64.32 | 153,686,279 |

**Table 3 Estimates of water hardship in the United States.**

| Percent of permittees listed as CWA Significant Noncomplier | Total | Mean | SD | Min | Max | N(counties) |
|---|---|---|---|---|---|---|
| Full EPA data | 6.01 | 9.00 | 12.54 | 0 | 100 | 3207 |
| Data duplicated with top and bottom 20% of counties | 3.87 | 7.16 | 9.85 | 0 | 100 | 3153 |

**Table 4 Estimates of elevated levels of water hardship in the United States.**

| Counties with greater than one percent of CWA permittees listed as CWA Significant Noncomplier | N(counties) | Percent | Population |
|---|---|---|---|
| Full EPA data | 2178 | 67.91 | 217,435,372 |
| Data duplicated with top and bottom 20% of counties | 1655 | 59.81 | 178,919,721 |

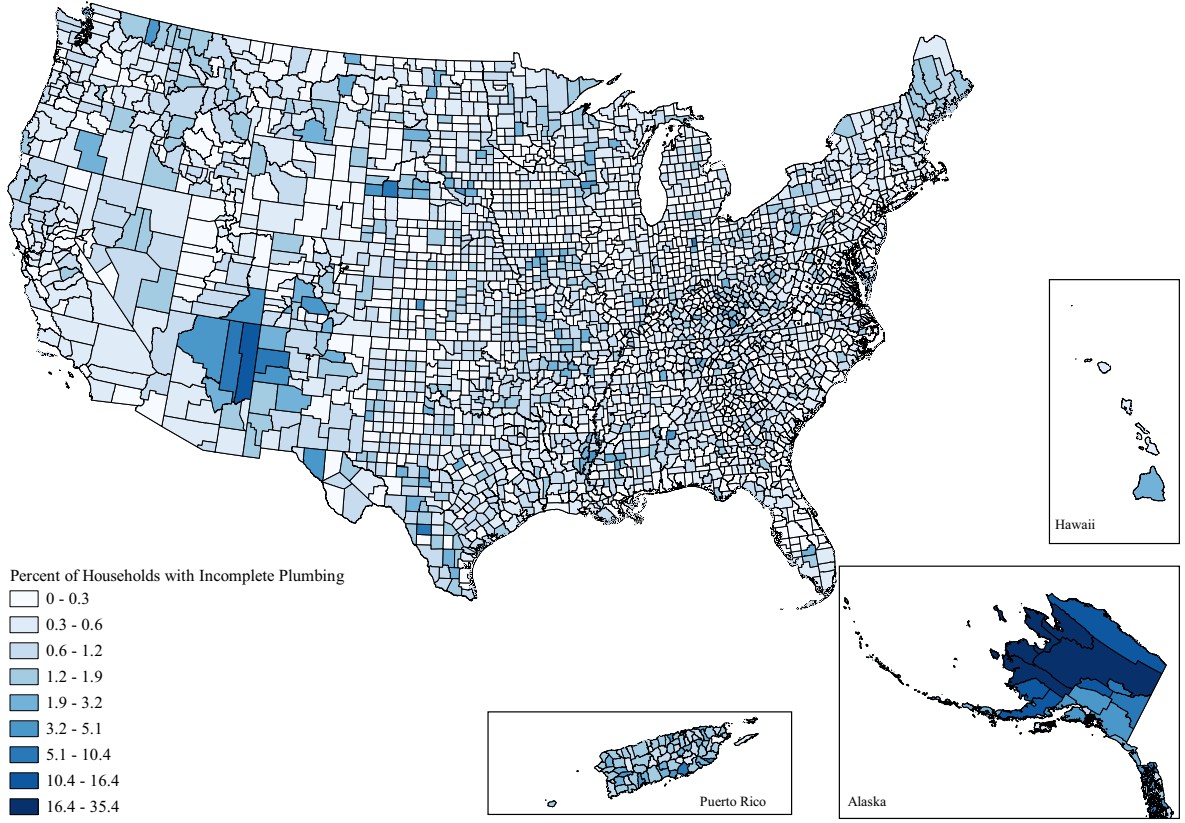

**Fig. 1 Map of the percent of county households without full indoor plumbing as reported by the 2014–2018 American Community Survey.** Households are determined to have incomplete plumbing if they do not have access to hot and cold water, a sink with a faucet, a bath or shower, and—up until 2016—a flush toilet.

Americans live in counties where >1% of community water systems are listed as Significant Violators, and more than 153 million Americans in the 39 reliable states and territories live in counties where greater than one percent of Clean Water Act permittees are Significant Noncompliers. Thus, although the number of individuals impacted by these issues is certainly far smaller than these totals, a vast number of Americans live in communities where issues of water quality are elevated.

Due to our conservative approach of removing all states with Clean Water Act data issues, we test the sensitivity of our estimates by also calculating supplemental estimates of Clean Water Act Significant Noncompliance under two counterfactual scenarios. In the first, we include the data as-is from the EPA for all counties in the 50 states, DC, and Puerto Rico, and in the second, we duplicate the counties in the top and bottom 20% of Significant Noncompliance in states without data issues—with the rationale being that the 945 counties removed due to poor data represented roughly 40% of the total counties remaining when problems states were removed. Thus, this attempts to simulate total counts if those removed were balanced between very high and very low levels of noncompliance. Results using all EPA data increase national estimates of Significant Noncompliance (Tables 3 and 4), with the total percent of permittees in this status jumping

from 3.37% to 6.01%. While the duplication test does raise our estimates, it is not nearly as dramatic, with the percent of permittees in Significant Noncompliance only rising to 3.87%. These results make sense given that the most common reason for data issues was an overreporting of noncompliance within states.

When looking at the issue spatially, we can see that while water hardship affects all parts of the country to some degree, the issues are clustered in space (Figs. 1–3). Importantly, the clustering varies between each water issue. Incomplete plumbing is clustered in the Four Corners, Alaska, Puerto Rico, the borderlands of Texas, and parts of Appalachia (Fig. 1); Safe Drinking Water Act Serious Violators are clustered in Appalachia, New Mexico, Alaska, Puerto Rico, and the Northern Intermountain West (Fig. 2); and Clean Water Act Significant Noncompliance clearly follows state boundaries—likely speaking to variable monitoring by state. Although spatial representation is limited by the absence of 13 states with inaccurate EPA data, we can still see that Clean Water Act Significant Noncompliance is clustered in the Intermountain West, the Upper Midwest, Appalachia, and the lower Mississippi (Fig. 3). These regional clusters persist when we include the problem states, which is visible in the map included in the Supplemental Information (Supplementary Figure 1).

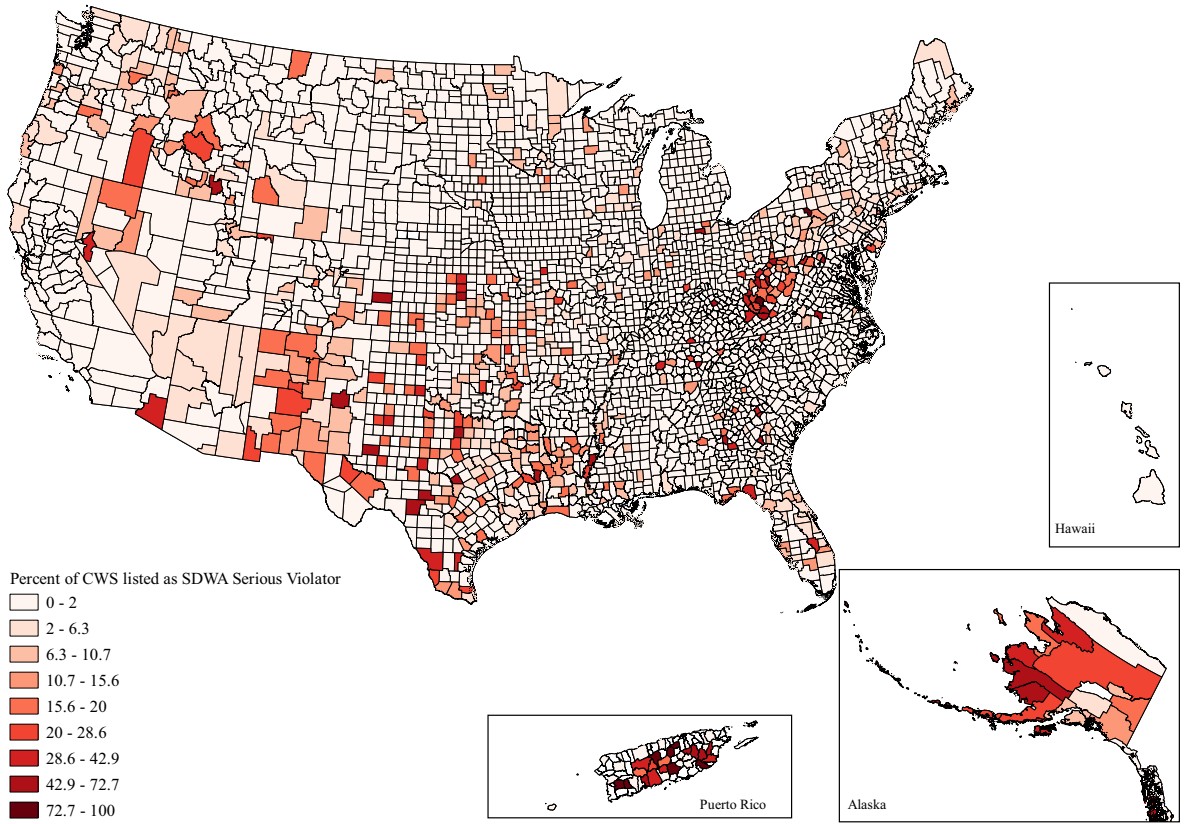

**Fig. 2 Map of the percent of active county community water systems listed as Safe Drinking Water Act (SDWA) Serious Violators.** Safe Drinking Water Act Serious Violators are those community water systems regarded by the Environmental Protection Agency as the most problematic due to violation and compliance history.

**Water injustice modeling**. Although we can easily see clustering by space in Figs. 1 through 3, the maps do not tell us whether or not incomplete plumbing and poor water quality are also clustered by social dimensions, which would represent an environmental injustice. To assess this social clustering, we estimate linear probability models of elevated levels of incomplete plumbing and poor water quality with the previously identified environmental justice dimensions of age, income, poverty, race, ethnicity, education, and rurality as our independent variables. We include these independent variables due to their prevalence within prior work on environmental injustice in both rural and urban areas[17–25]. Further, although there is not a one-to-one overlap, these variables conceptually map onto the dimensions of the Center for Disease Control Social Vulnerability Index: Socioeconomic Status (i.e. income, poverty, education), Household Composition & Disability (i.e. age), Minority Status & Language (i.e. race and ethnicity), and Housing & Transportation (i.e. rurality)[28].

For each outcome, we first estimate purely descriptive models with only one dimension of injustice included at a time, and then estimate a full model with all dimensions included. The outcomes are dichotomous measures of whether or not a county had >1% of households with incomplete plumbing, >1% of community water systems listed as Serious Violators, or >1% of Clean Water Act permittees in Significant Noncompliance. All descriptive statistics for the dichotomous outcomes are presented in Table 2. Descriptive statistics for the continuous independent variables are presented in Supplementary Information (Supplementary Table 1). Here we present the outcomes of the purely descriptive models visually in Fig. 4 and discuss the full models in the narrative. Full regression results, including exact 95% confidence intervals and *p*-values, for all models are available in Supplementary Information (Supplementary Tables 2, 3 and 4).

We find elevated levels of incomplete plumbing at the county level were significantly ($p < 0.05$) associated with older populations, lower income, higher poverty, greater portions of indigenous people (American Indian, Alaska Natives, Native Hawaiian, and Other Pacific Islanders), lower levels of education, and more rural counties (Fig. 4). A great deal of these associations persisted in a full model with all dimensions of injustice (Supplementary Table 2). The only differences between the full model and the series of purely descriptive models were that income, percent with at least a bachelor's degree, and

non-metropolitan metropolitan adjacency were no longer significantly associated with elevated levels of incomplete plumbing. This indicates that the inequalities in plumbing access along the dimensions of age, poverty, indigeneity, low education, and extreme rurality persist at the county level, even when accounting for the other dimensions of environmental injustice.

The models for elevated levels of Safe Drinking Water Act Serious Violators indicated less social inequality than the models for incomplete plumbing. The purely descriptive models found elevated levels of Serious Violators were associated with higher income, higher poverty, and metropolitan counties (Fig. 4). The full model had minor variation, with median household income no longer being significant in the model (Supplementary Table 3). Thus, the full model shows that the association between elevated levels of Serious Violators and higher poverty and metropolitan status persists even when considering other social dimensions.

We see the fewest indicators of water injustice for elevated levels of Clean Water Act Significant Noncompliance—which only include counties within the 39 states and territories with accurate data. In the purely descriptive models, we find older populations, more Latino/a counties, less educated counties, and remote rural counties were significant less likely to have elevated levels of noncompliance (Fig. 4). In the full model, the association for education is no longer significant but age, Latino/a, and rurality remain (Supplementary Table 4). Similar to our national estimates, we also conducted model sensitivity tests using the same scenarios described above. As shown in Fig. 5, neither scenario substantively changes our conclusions, with the only changes in significance being for percent Latino/a and percent without a high school diploma—both of which were only marginally significant in our primary models ($p > 0.01$).

## Discussion

Our findings demonstrate that the problem of water hardship in the United States is hidden, but not rare. Indeed, millions live in counties where more than 1 out of 100 occupied households lack complete plumbing. Millions more live in places with chronic Safe Drinking Water Act violations and Clean Water Act noncompliance. We present this paper to help sound the alarm of this significant household water crisis in the United States. Although the relative share

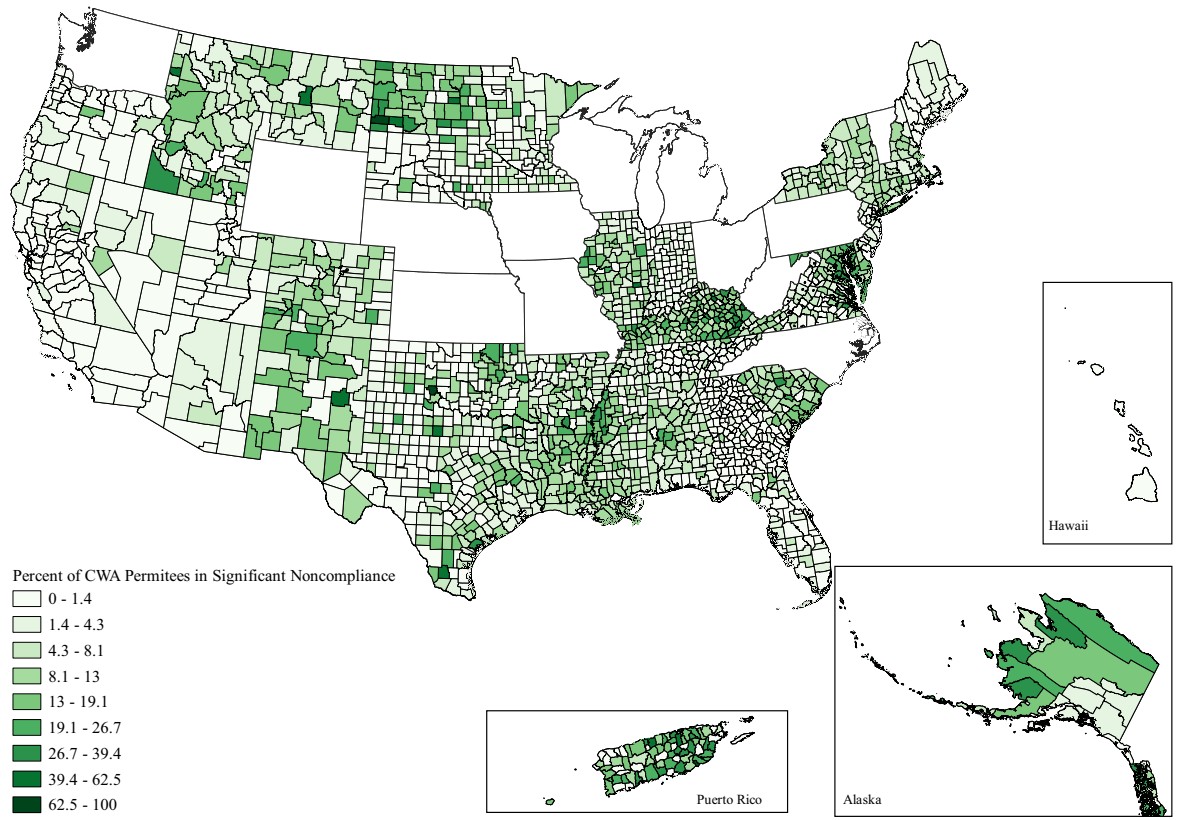

**Fig. 3 Map of the percent of county Clean Water Act (CWA) permittees listed as Clean Water Act Significant Noncompliers.** All facilities that discharge directly into water of the United States are issued a Clean Water Act permit, those who represent a more severe level of environmental threat due to violations and noncompliance are considered in Significant Noncompliance.

of Americans experiencing this problem is low, the absolute number of people dealing with incomplete plumbing—a total of 489,836 households—and poor water quality—1165 community water systems nationwide and 9457 Clean Water Act permittees in the 39 accurate states and territories—remains quite high. Further, given the water infrastructure of the United States, consistently deemed as poor by experts[6,11], if action is not taken the situation may only get worse.

These findings are even more concerning when considering that water hardship is spread unevenly across both space and society, reflecting the spatial patterning of social inequality due to settler colonialism, racism, and economic inequality in the United States. Figures 1, 2, and 3 document the clear regional clustering of these issues and our models of environmental injustice demonstrate the social inequalities found for this form of hardship. Particularly in the case of incomplete plumbing, we find significant environmental injustice at the county level along the social dimensions of age, income, poverty, indigeneity, education, and rurality. These associations certainly stem from multiple causal pathways—for example associations with indigeneity likely stem from legacies of injustice as well as ongoing policies placing limitations on land use and infrastructure development on American Indian reservations[15]. Remedying these injustices will require careful attention to the root causes of the problem. It is important to note that the signs of injustice for poor water quality were less clear than for incomplete plumbing, with far fewer significant associations. Further, the minimal support for injustice in the case of Clean Water Act Significant Noncompliance was evident in all three specifications of counties in our sensitivity tests. Suggesting that the removal of the states with data issues did little to impact coefficient estimates. These differences between dimensions of water hardship highlight the nuance between each of these specific forms of water hardship, and suggest a one-size-fits-all approach to remedying this crisis is unlikely to be effective. This need for place-based policy is made stark when we view the obvious state level differences in Clean Water Act Significant Noncompliance in Fig. 3. A clear direction for future work is to investigate the cause of these notable state-level differences.

The household water access and quality crisis we have identified here is solvable. Policy is needed to specifically address these issues and bring this problem into the spotlight. However, as indicated by the persistently high levels of Safe Drinking Water Act Serious Violation and Clean Water Act Significant

Noncompliance, any policy put in place must be enforceable and strong. As it currently stands, counties with elevated levels of incomplete plumbing and poor water quality in America—which are variously likely to be more indigenous, less educated, older, and poorer—are continuing to slip through the cracks.

## Methods

**Data sources**. Data for this analysis were extracted from the American Community Survey (ACS) 5-year estimates for 2014–2018 via Integrated Public Use Microdata Series – National Historic Geographic information System (IPUMS-NHGIS)[26], and from the Environmental Protection Agency's (EPA) Enforcement and Compliance History Online (ECHO) Exporter[27]. Data were extracted at the county level for all 50 states, Washington DC, and Puerto Rico–the two non-state entities with available data. The ACS is an ongoing survey of the United States which documents a wide variety of social statistics ranging from simple population counts to housing characteristics. Due to the staggered sampling structure of the ACS, it takes 5 years for every county to be sampled. Because of this, researchers must use 5-year intervals to ensure complete data coverage. The data from these 5 years are projected into estimates for all counties in the United States for the 5-year period in question. As of this study, 2014–2018 was the most recently available data.

ECHO collates data from EPA-regulated facilities across the United States of America to report compliance, violation, and penalty information for all facilities for the most recent 5-year interval. ECHO data is updated weekly and the data for this paper was extracted on 18 August 2020. This means that the data in our analysis represents the status of each community water system or Clean Water Act permittee, as reported by the EPA, as of 18 August 2020. Only those community water systems or Clean Water Act permittees listed as Active by ECHO were included in this analysis. As ECHO data is at the level of the water system, permittee, or utility, we aggregated data up to the county level.

Safe Drinking Water Act data was geolocated using QGIS 3.10 based upon latitude and longitude. This was done because other geographic identifiers for the Safe Drinking Water Act data were often missing. In line with prior work[4,5,7,8], and in order to facilitate a cleaner dataset, we only focus on those water systems labeled community water systems for our analysis. Community water systems were geolocated based upon the county in which their latitude and longitude were located, if a community water system had latitude and longitude over water, a nearest neighbor join was used. In total, 1334 out of 49,479 community water systems were dropped because of there being no reported latitude or longitude. Of these, a total of 4.0%, or 54 community waters systems, were reported as in serious violation. It should be noted that the EPA is aware of a small number of water systems in Washington for which ECHO data may be inaccurate. However, since this is a small number and it is not listed as a 'Primary Data Alert,' we retain all states in this portion of the analysis. Finally, the EPA is generally aware that there are "inaccuracies and underreporting of some data in this

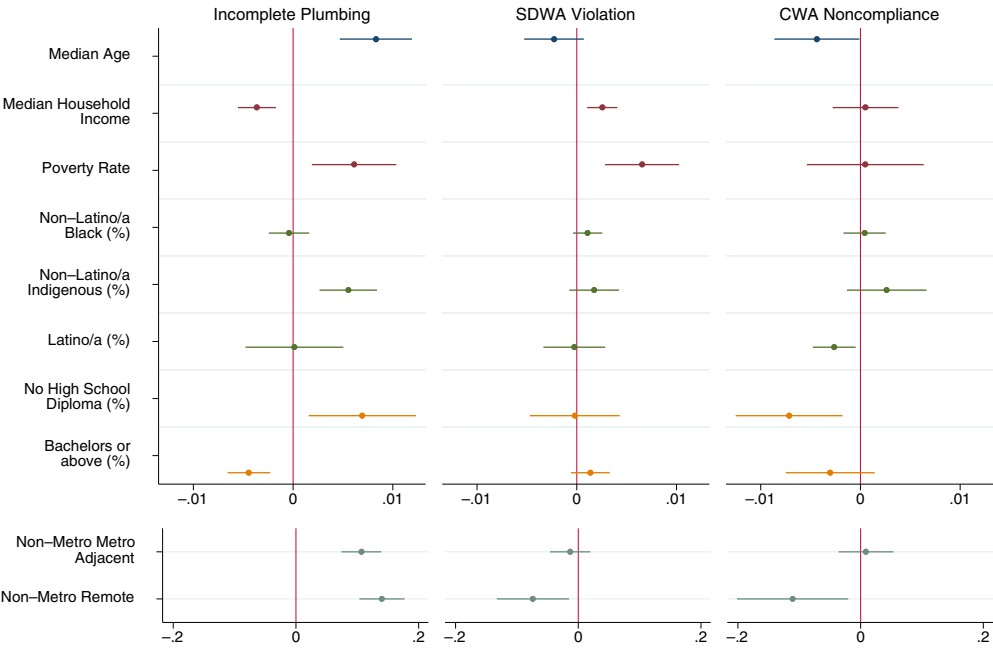

**Fig. 4 Coefficient plot of descriptive regression model results.** Different colors for plotted coefficients represent separate blocks of variables. Models are linear probability models with state fixed effects and cluster-robust standard errors at the state level. All tests two-tailed. Dots indicate point estimates and lines represent 95% confidence intervals. Models predicted elevated levels of each dimension of water hardship. For incomplete plumbing this is indicated by >1% of households in a county having incomplete plumbing ($N = 3219$). For Safe Drinking Water Act (SDWA) Serious Violation this is indicated by >1% of active community water systems being considered Serious Violators ($N = 3143$). For Clean Water Act (CWA) Significant Non-Compliance this is indicated by >1% of Clean Water Act permittees being considered in Significant Non-Compliance ($N = 2261$). Full model results, confidence intervals, and exact p-values available in SI.

system," which is listed as a Primary Data Alert[27]. However, due to the lack of specifics, we cannot exclude inaccurate cases. Thus, our analysis should be viewed as reflecting drinking water quality is *as reported by the EPA* in August of 2020, which may reflect some level of inaccuracy.

Active Clean Water Act permittees were first identified by listed county. This was done because 345,176 out of 350,476 permittees had a county reported. Those without a county reported were located using latitude and longitude in the same manner as community water systems. There were 10 permittees without latitude and longitude or county listed which were excluded from our analysis. Of these, seven were in significant noncompliance and three were not. Due to some Clean Water Act permittees having latitude and longitude placements far away from the United States, those over 100 km from their nearest county were excluded from analysis. Unfortunately, ECHO data for the Clean Water Act data during the study period is inaccurate for 13 states. Although the nature of the inaccuracy varies from state to state, these issues generally stem from difficulties in transferring state data into the federal system. Due to this, these states appear to have far more permittees in Significant Noncompliance than are actually in violation. To address this issue, we removed all counties within these states from our Clean Water Act analysis. The impacted states include Iowa, Kansas, Michigan, Missouri, Nebraska, North Carolina, Ohio, Pennsylvania, Vermont, Washington, West Virginia, Wisconsin, and Wyoming[29]. Finally, for community water systems and Clean Water Act permittees, some counties (76 for community water systems and 5 for Clean Water Act permittees) had no reported cases. Those counties were treated as zeroes for cartography and as missing for modeling purposes.

Similar to prior work in this area[4,5,8], we restrict our analysis to the scale of the county for reasons related to data limitations and resulting conceptual validity. Although counties are arguably larger in geographic area than ideal for an environmental injustice analysis, if we were to use a smaller unit for which data is available such as the census tract, the conceptual validity of the analysis would be limited due to the apolitical nature of these units. As outlined above, ECHO data is messy and missing many geographic identifiers. What is provided is generally either the county or latitude and longitude. If only the county is provided, then we are constrained to using the county regardless of conceptual validity. However, even when latitude and longitude are provided—which is the case for many observations—the provided point location says nothing about which households the water system or permittee serves or impacts. Due to this, whatever geographic unit we use carries the assumption that those in the unit could be plausibly impacted by the water system or permittee. Given that counties are often responsible for both regulating drinking water, as well as maintaining and providing water infrastructure[30], we were comfortable with this assumption between point location and presumed spatial impact when using the scale of the county. However, we believe this assumption would have been invalid and untestable for smaller apolitical units for which demographic data is available such as census tracts.

Beyond the issues presented by ECHO data, the county is also the appropriate scale of analysis for this study due to the estimate-based nature of the ACS. ACS estimates are based on a rolling 5-year sample structure and often have very large margins of error. At the census tract level, these standard errors can be massive, especially in rural areas[31–33]. Due to this variation, and the need to include all rural areas in this analysis, the county, where the margins of error are considerably smaller, is the appropriate unit for this study. All of this said, the county is, in fact, a larger unit than often desired or used in environmental justice studies.

Studies focused on exclusively urban areas with clearer pathways of impact can and should use smaller units such as census tracts. It will be imperative for future scholarship focused on water hardship across the rural-urban continuum to gain access to reliable data on sub-county political units, as well as data linking water systems to users, to continue documenting and pushing for water justice.

**Dependent variables.** The dependent variables for this analysis were assessed in both a continuous and dichotomous format. For descriptive results and mapping, continuous measures were used. For models of water injustice, a dichotomous measure which classified counties as either having low levels of the specific water issue or elevated levels of the specific water issue, was used due to the low relative frequency of water access and quality issues relative to the whole United States population. For all three outcomes, we benchmark an elevated level of the issue as what would be viewed as an unacceptable level under United Nations Sustainable Development Goal 6.1, which states, "by 2030 achieve universal and equitable access to safe and affordable drinking water for all"[1]. As this goal focuses on ensuring all people have safe water, we deem a county as having an elevated level of the issue if >1% of households, community water systems, or permittees had incomplete plumbing, were in Significant Violation, or Significant Noncompliance, respectively. Although we could have used an even stricter threshold given the SDG's emphasis on ensuring access for all people, we use 1% as our cut-off due to its nominal value and ease of interpretation.

For water access, the continuous measure was the percent of households in a county with incomplete household plumbing as reported by the ACS. The ACS currently asks respondents if they have access to hot and cold water, a sink with a faucet, and a bath or shower. Up until 2016, the question also included a flush toilet[34]. As we must use the most recent 2014–2018 5-year estimates to establish full coverage of all counties, this means that incomplete plumbing in this item may, or may not include a flush toilet depending on when the specific county was sampled. The dichotomous version of this variable benchmarked elevated levels of incomplete plumbing as whether or not 1% or more of households in a county had incomplete plumbing.

Water quality was assessed via both community water systems from the Safe Drinking Water Act, and from permit data via the Clean Water Act. For Safe Drinking Water Act data, the continuous measure was the percent of community water systems within a county classified as a Safe Drinking Water Act Serious Violator at time of data extraction. The EPA assigns point values of either 1, 5, or 10 based upon the severity of violations of the Safe Drinking Water Act. A Serious Violator is one who has "an aggregate score of at least eleven points as a result of some combination of: unresolved more serious violations (such as maximum contaminant level violations related to acute contaminants), multiple violations (health-based, monitoring and reporting, public notification and/or other violations), and/or continuing violations"[27]. The dichotomous measure benchmarked elevated rates of Safe Drinking Water Act Significant Violation as whether or not >1% of county community water systems were classified as Serious Violators.

For Clean Water Act permit data, the continuous measure was the percent of permit holders listed as in Significant Noncompliance at the time of data extraction. Significant Noncompliance in the Clean Water Act refers to those permit holders who may pose a "more severe level of environmental threat" and is based upon both pollution levels and reporting compliance[27]. The dichotomous measure again set the threshold for elevated levels of poor

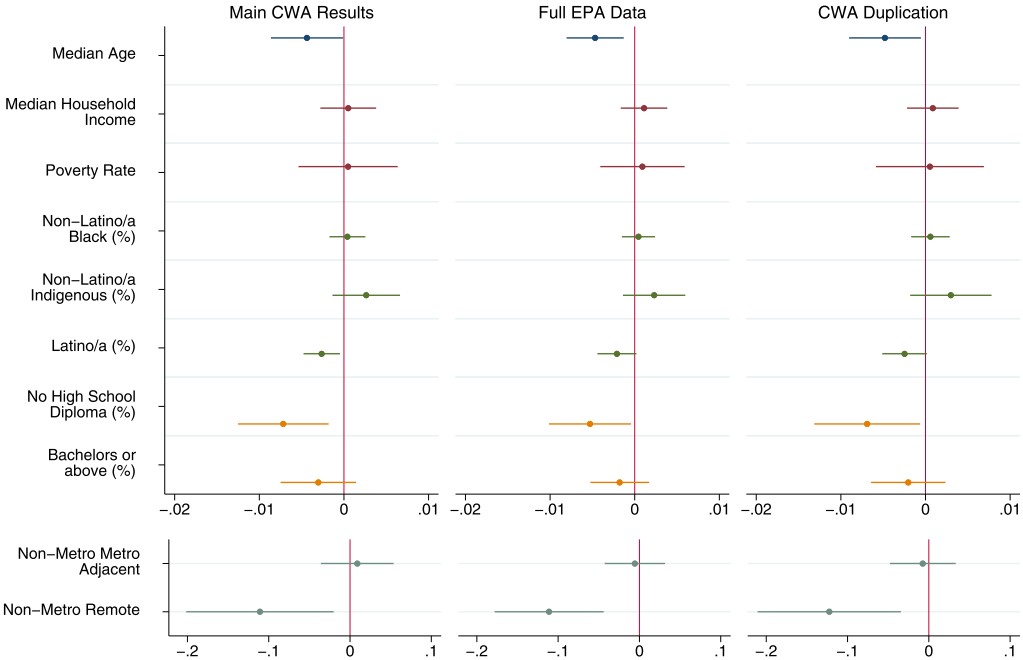

**Fig. 5 Coefficient plot of Clean Water Act sensitivity test results.** Descriptive regression model results. Different colors for plotted coefficients represent separate blocks of variables. Models are linear probability models with state fixed effects and Huber/White/Sandwich cluster-robust standard errors at the state level. All tests are two-tailed. Dots indicate point estimates and lines represent 95% confidence intervals. Models predicted whether or not there were greater than 1% of Clean Water Act permittees being considered in Significant Noncompliance in the county. First model excludes counties in states with CWA data issues (N = 2261), second model includes all counties reported by the EPA (N = 3206), third model duplicates counties in the top and bottom 10% of CWA Significant Noncompliance within states without data issues (N = 3151). Full model results, confidence intervals, and exact p values available in SI.

water quality at whether or not >1% of Clean Water Act permittees in a county were listed as in Significant Noncompliance at time of data extraction.

**Independent variables**. The independent variables we include in models of water injustice are those frequently shown to be related to environmental injustice in the United States. These include age, income, poverty, race, ethnicity, education, and rurality[17–25]. Age was included as median age. Income was included as median household income. Poverty was the poverty rate of the county as determined by the official poverty measure of the United States[35]. Race and ethnicity was included as percent non-Latino/a Black, percent non-Latino/a indigenous, and percent Latino/a. Because the focus was on indigeneity, percent American Indian or Alaska Native was collapsed with Native Hawaiian or Other Pacific Islander. We did not include percent non-Latino/a white due to issues of multicollinearity. Finally, rurality was included as a three-category county indicator of metropolitan, non-metropolitan metropolitan-adjacent, and non-metropolitan remote, as determined by the Office of Management and Budget in 2010[36]. The OMB determines a county is metropolitan if it has a core urban area of 50,000 or more people, or is connected to a core metropolitan county by a 25% or greater share of commuting[36]. A non-metropolitan county is simply any county not classified as metropolitan. Non-metropolitan metropolitan adjacent counties are those which immediately border a metropolitan county, and non-metropolitan remote counties are those that do not.

**Water injustice modeling approach**. Water injustice was assessed by estimating linear probability models for the three dichotomous outcome variables with state fixed effects to control for the visible state level heterogeneity and differences in policy, reporting, and enforcement (e.g. the clear state boundary effects in Fig. 3). We employ the conventional Huber/White/Sandwich cluster-robust standard errors at the state level—which account for heteroskedasticity while also producing a consistent standard error estimate in-light of the lack of independence found between counties in the same state. All modeling was performed in Stata 16.0 and mapping was performed in QGIS 3.10. We assessed all full models for multicollinearity via condition index and VIF values and the independent variables had an acceptable condition index of 5.48 for incomplete plumbing and Safe Drinking Water Act models and 5.63 for Clean Water Act models, well below the conservative cut-off of 15, as well as VIF values of <10. We initially included percent non-Latino/a white as an independent variable, but removed the item due to unacceptably high condition index levels (>20). All indications of statistical significance are at the $p < 0.05$ level and 95% confidence intervals and exact p-values of all estimates are provided in Supplementary Information. Each dependent variable was analyzed through a series of six models. First, we estimated separate purely descriptive models, where the only independent variables included were those associated with that specific dimension and the state fixed effects, for all five dimensions of environmental injustice. After estimating these five models, we estimated a full model including all social dimensions at once.

The reason for this approach was to ensure that we provided a robust descriptive understanding of the on-the-ground social patterns of water hardship, in addition to a full model showing the strongest social correlates of this issue. For example, if when we only included income variables we found that incomplete plumbing is less likely in counties with

higher median incomes, but this effect goes away when we include other social variables, this does not remove the fact that there is an unequal distribution of incomplete plumbing by income on-the-ground. All that it means is that this income effect does not persist over and above the other social dimensions of environmental injustice. It may be that once other dimensions such as structural racism, captured by race and ethnicity variables, are considered, income is no longer a significant predictor. However, at a pure associational level, incomplete plumbing would still be unequally distributed by income on-the-ground. In fact, this is exactly what we find for incomplete plumbing (Supplementary Table 2). Due to this, both the pure descriptive and full models are needed for full understanding. Complete tables of all results are presented in the Supplementary Information File (Supplementary Tables 1 through 4).

**Sensitivity tests**. Due to our conservative approach to remove all problem states from the Clean Water Act portion of our analysis, we conducted a series of sensitivity tests wherein we generated national estimates of Significant Noncompliance, as well as models of elevated Significant Noncompliance under two scenarios (Supplementary Tables 5 and 6). In the first scenario we include all data reported by the EPA, meaning that we use all data for the 50 states, DC, and Puerto Rico, regardless of any EPA data flags. In the second scenario, we replaced the data lost when dropping states by duplicating the counties in the top and bottom 20% of significant violations in the remaining counties. The top and bottom 20% was chosen because the 945 counties removed when the 13 states were dropped was roughly equal to 40% of the remaining 2262 counties. This counterfactual allows us to get closer to a plausible estimate of the absolute scope of CWA Significant Noncompliance by adopting a scenario where the counties dropped in problem states were either very high, or very low in terms of Significant Noncompliance. Functionally, duplicating the bottom 20% posed a challenge because the bottom 30% of counties had zero permittees in Significant Noncompliance. This zero-bias is one of the primary reasons why our outcome variable was dichotomized. To address this, we randomly selected two-thirds of these counties for duplication using a seeded pseudorandom number generator in Stata. Following duplication of cases, all estimates and models were generated in the same manner as the primary models of this study.

**Reporting summary**. Further information on research design is available in the Nature Research Reporting Summary linked to this article.

## Data availability
The raw and geolocated datasets are publicly available on the Open Science Framework project for this study at https://doi.org/10.17605/OSF.IO/ZPQR9 (https://osf.io/zpqr9/).

## Code availability
Analysis code is available on the Open Science Framework project for this study at https://doi.org/10.17605/OSF.IO/ZPQR9 (https://osf.io/zpqr9/). As the raw data was not geolocated using a code-based operation, code for this portion of the analysis is not

available. However, the raw data is posted, and should researchers wish they will be able to use our description provided here to replicate geolocation using the GIS software of their choice. All other elements of the analysis are easily replicated via our provided code. As the both the raw and geolocated datasets are provided, replication of our analysis should be straightforward.

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

## Acknowledgements

The authors would like to acknowledge Tom Dietz, Lauren Mullenbach, Matthew Brooks, and Jan Beecher for their feedback on this manuscript. They would also like to thank Colleen Keltz at the Washington State Department of Ecology for alerting us to the issues with Clean Water Act data for Washington and other states.

## Author contributions

Conceptualization: J.T.M. and S.G.; methodology: J.T.M.; formal analysis: J.T.M.; data curation: J.T.M.; writing- original draft preparation: J.T.M. and S.G.; writing – review and editing: J.T.M. and S.G.; visualization: J.T.M.

## Competing interests

The authors declare no competing interests.
