## [Peer Review File · Nature Communications]

Reviewer comments, first round

Reviewer #1 (Remarks to the Author):

This manuscript presents a quantitative case study that utilizes the distributive environmental justice (EJ) framework to conduct county-level analysis of social inequalities in the distribution of incomplete plumbing and poor water quality in the U.S. The topic is timely, relevant, and important, and the paper is generally well-written and well-organized. The conceptual framework and literature review are both appropriate, and the results provide statistical evidence of greater water hardship in counties containing higher proportions of specific socially vulnerable groups. However, there are several methodological weaknesses that render the statistical findings somewhat questionable and make this paper unsuitable for publication in its current form. My main concerns are described below:

1-Counties are not appropriate spatial units for the type of distributive EJ analysis presented in this paper, and no theoretical or empirical justification is included for using counties as units of analysis. Previous EJ studies suggest that county boundaries are too large in size and counties are too coarse in scale to represent impacted communities. I strongly recommend the use of smaller analytic units such as census tracts. Almost all distributive EJ studies conducted in the last 15 years that are national in scope, including those cited in this paper (United Church of Christ 1987; Mohai and Saha 2006, 2015), have all relied on units smaller than counties. Additionally, the use of larger areal units such as counties have been found to increase the strength and significance of statistical associations between risk indicators and socio-demographic variables (when compared to use of smaller units such as tracts) in previous EJ research.

2-This study fails to acknowledge the significant data quality problems that have been identified in previous socio-demographic research on the ACS. Recent studies illustrate how the ACS 5-year average estimates can have huge margins of error when compared to the 2010 census (e.g., Bazuin and Fraser, 2013; Folch et al., 2016). To mitigate against measurement error, Folch et al. (2016) suggested using ACS estimates whose margin of error was lower than half of the estimate—that is, where the coefficient of variation (CV) for the estimate was less than 0.40. Several studies have also tried to derive reliable proportional estimates by excluding census units with small population and/or housing counties from their analysis. Although this study utilizes county-level ACS estimates, the authors do not acknowledge any data quality problems with the ACS socio-demographic variables and how they might have been mitigated against with appropriate techniques.

3-Although multivariate regression models are presented and used to draw important conclusions in this study, there is no discussion or examination of multicollinearity. Since most of independent variables in the models presented in Table 2, 3, and 4 are highly likely to be strongly correlated with each other (e.g., median household income vs. poverty rate) at the county level, it is important to analyze this issue and indicate how it was addressed for the multivariate analysis.

4-More justification is necessary on the explanatory variables chosen to measure the distributive EJ implications. The authors should consider using variables that comprise the widely used and CDC-recommended social vulnerability index (Flanagan et al. 2018: <https://www.ncbi.nlm.nih.gov/pmc/articles/PMC7179070/>), instead of using an arbitrary subset of these variables. It is important to at least consider how the selected variables represent the domains established there.

5- On a related note, I found some of the choices for explanatory variables (e.g., % Black and % White) to be problematic. Since the White and Black categories from the U.S. Census or ACS include both Hispanics and non-Hispanics, only % non-Hispanic White and % non-Hispanic Black should have been used to avoid double counting problems. This would also make the statistical results consistent and comparable with most recent national scale distributive EJ studies conducted

in the U.S.

Reviewer #2 (Remarks to the Author):

This is a fantastic and very much needed overview of the household water crisis in US. It is clearly written and very accessible. The most noteworthy results stem from the spatial patterning of different forms of water hardship and how these differences are unlikely to be resolved with a “one size fits all” approach. The manuscript is well situated within the literature to make significant contributions around water security, environmental justice, sustainability planning, and related fields. The methods appear robust, are clearly outlined and replicable, and I cannot discern any major flaws within the analysis.

I only have a couple of minor quibbles with the manuscript, which center around the need to make a few points more clear. The first is on page 2, line 72, when the authors state that “much of the work has become dated and it generally focuses on a single dimension of the issue at a time”, can they provide an example? Sure, some work is dated, but what are examples of the single dimension problem?

Second, it seems like Indigenous/non-indigenous space matters in how the household water crisis takes spatial form. A more critical and/or explicit take could be made here about longstanding injustices of settler colonialism, etc. A good place to do this might be around page 3, line 82 or page 5, line 183. I think only a sentence or two somewhere would suffice.

Finally, and I am not sure if there is space within the manuscript to do this, but Figure 3 is quite glaring in terms of state differences. I see the authors note that it likely speaks to variable monitoring by state, but that can't be all that explains that, can it? It's incredible how much some of the states (WA, WY, WV, IA, MO, WI) stand out. They're all quite different, so it really begs the question for me around what is shared among these states. But it also raises the question of the meaningfulness of this measure, if it's only monitoring leading to high rates of non-compliance. Would there be higher rates of non-compliance everywhere with higher rates of monitoring? Anyways, I think some of these type of questions could be cleared up somewhat in the MS or added to the methods.

Generally, I think this is a great manuscript and I can't wait to use it for teaching and research purposes. It will be a great resource. I see it nearly publishable as is and am very thankful for the opportunity to review it.

Reviewer #3 (Remarks to the Author):

This paper investigates the prevalence of access to clean drinking water in the United States. The authors use EPA reports of water quality violations and ACS estimates of complete plumbing as measures of water quality and plumbing respectively. They subsequently perform a statistical analysis to determine any socio-demographic factors which associate with clean water access issues.

Overall, the paper is well written, clear, and methodologically sound. I have a few larger comments in the interest of strengthening the manuscript and minor details.

Major comments (in the order they arise when reading):

My first major comment is about the framing of the study. From the summary and the introductory body of text, the results pertaining to the scope of the lack of complete plumbing are sold as the novel aspect of the study (Lines 37-40 and 53-57 for example). This is just reporting the data

provided by the ACS and—in the opinion of this reviewer— does not represent the contribution of the paper.

I ask that either appropriate language be included to make it clear this data already exists (e.g. “Data from the American Community Survey shows XYZ houses are without access to complete plumbing...”) or more favorably, the introductory sections be framed to highlight other aspects of the work such as the factors associated with clean water violations or complete plumbing access.

My second major comment is about 1/100 cutoff used to develop the dichotomous statistical models. I don’t disagree with this notionally, but further justification is needed to pick 1/100 as a cutoff. One route would be to show the distribution of the county-level fraction of the population with incomplete access to plumbing and show where 1% falls in that distribution. From a quick look at ACS data, 2757 counties have between 0-1% of the household lacking plumbing with 462 above that threshold. Another route would be to compare the results of the statistical model at multiple thresholds.

My third major comment is about the use of population-level statistics. In lines 106-107 and 123-126, the statistics presented are the total population residing in counties in which over 1% of the population is in water crisis. This is borderline-misleading the way it is presented, as it would be easy to mistake this for the total population that did not have indoor plumbing. This is more an issue when used in lines 123-126, as the results moves from percentages directly to the ‘fraction in the county’ numbers.

I understand the utility of this measurement as a way to describe the prevalence of this issue geographically (i.e. saying this is something that could be occurring in your community). However, further clarification is needed, preferably by stating clearly a percentage of the population impacted followed by the absolute number impacted and moving this type of broader impact statistic to the discussion.

Minor comments:

With American Indian and Alaskan Native populations found to be more likely to have incomplete plumbing and incomplete water, there may be more at play as a result of federal policies which place significant limitations on what development can be done on native lands and may contribute to lack of access to plumbing.

Line 118: is it possible to determine the overlap between CWA noncompliant systems and SDWA violators?

Lines 331-332: Methodologically, I fully understand why dropping data without reported lat/lon is a reasonable decision. I worry, however, that there may be association between missing data and noncompliant facilities. Poor management or lack of institutional knowledge of EPA regulations and compliance immediately come up as reasons why a facility may be both non compliant and reporting incomplete data. If feasible, this reviewer would like to see a brief summary of the rate of compliance of facilities which were dropped (both for SDWA and CWA). If they are atypically high, this furthers the case of the paper that this is an under-reported issue. If they low, this should be included as a discussion point.

Figures 1,2 and 3 are difficult to assess trends from in areas dense with counties. I suggest the following changes

(1) use a lighter color for the county borders so areas like northeastern TN, western NC, central KY and southern IN are more easily identified. This should also help the visual congestion in coastal areas.

(2) Use a consistent scale throughout. I’m unsure if it’s an issue with a map projection or image size, but currently Puerto Rico looks to be about the size of Mississippi which is certainly not true in reality

(3) Eastern Maine is cut off

Thank you to the authors for presenting a clear and purposeful manuscript.
Dr. Ben Rachunok

Reviewer 1	
Comments	Note
This manuscript presents a quantitative case study that utilizes the distributive environmental justice (EJ) framework to conduct county-level analysis of social inequalities in the distribution of incomplete plumbing and poor water quality in the U.S. The topic is timely, relevant, and important, and the paper is generally well-written and well-organized. The conceptual framework and literature review are both appropriate, and the results provide statistical evidence of greater water hardship in counties containing higher proportions of specific socially vulnerable groups. However, there are several methodological weaknesses that render the statistical findings somewhat questionable and make this paper unsuitable for publication in its current form. My main concerns are described below:	Thank you for your detailed review. We have responded to each of your comments below and hope you will now find the paper suitable for publication.
1-Counties are not appropriate spatial units for the type of distributive EJ analysis presented in this paper, and no theoretical or empirical justification is included for using counties as units of analysis. Previous EJ studies suggest that county boundaries are too large in size and counties are too coarse in scale to represent impacted communities. I strongly recommend the use of smaller analytic units such as census tracts. Almost all distributive EJ studies conducted in the last 15 years that are national in scope, including those cited in this paper (United Church of Christ 1987; Mohai and Saha 2006, 2015), have all relied on units smaller than counties. Additionally, the use of larger areal units such as counties have been found to increase	We apologize for a lack of clarity regarding the reason for our decision to use the spatial scale of county, as well as our lack of attention to difficulties in the ACS. Because this point is so important, and we did not choose to redo our analysis at the census tract level, we provide a more protracted justification here. Please note that both points 1 and 2 are addressed here because they are intimately related. To begin, we fully agree that the county is larger than desirable for assessing issues of Environmental Injustice. Counties vary in size across the country, can be very large, and display considerable internal heterogeneity. That said, in-

WATER HARDSHIP

the strength and significance of statistical associations between risk indicators and socio-demographic variables (when compared to use of smaller units such as tracts) in previous EJ research.

2-This study fails to acknowledge the significant data quality problems that have been identified in previous socio-demographic research on the ACS. Recent studies illustrate how the ACS 5-year average estimates can have huge margins of error when compared to the 2010 census (e.g., Bazuin and Fraser, 2013; Folch et al., 2016). To mitigate against measurement error, Folch et al. (2016) suggested using ACS estimates whose margin of error was lower than half of the estimate—that is, where the coefficient of variation (CV) for the estimate was less than 0.40. Several studies have also tried to derive reliable proportional estimates by excluding census units with small population and/or housing counties from their analysis. Although this study utilizes county-level ACS estimates, the authors do not acknowledge any data quality problems with the ACS socio-demographic variables and how they might have been mitigated against with appropriate techniques.

line with other research on this specific form of injustice (Allaire et al., 2018; Gasteyer et al., 2016; McDonald & Jones, 2018), retaining the unit of the county for this analysis is necessary for two reasons. 1) the reliability and availability of census data, and 2) the data available on water quality as it relates to political boundaries.

Regarding the first reason, ACS data for census tracts are unreliable, leading some to argue variable-based models of census tract data are inappropriate (Spielman & Singleton, 2015). You raise this point yourself in Point 2. However, in rural areas it is far worse, especially in low income areas (Folch, et al. 2016). The literature you cite discussing other approaches for using census tracts generally focuses on urban areas, where census tracts are a more plausible and usable spatial unit. This makes it impossible to conduct our analysis at the census tract level because rurality is a very important dimension of water hardship and we need to have as complete data as possible for our dependent variable. Further, we chose to use the county *a priori* because of the issues in the ACS you mention. While not perfect, ACS data at the county level is far more reliable and has much lower margins of error than census tract estimates. For example, we downloaded the equivalent census tract data for plumbing and found that at the county level the average ratio of margin of error to estimate for the number of owner occupied housing units with complete plumbing was 0.048, compared to the same statistic being 0.15 at the census tract level. This massive difference in precision is for the more common occurrence of complete plumbing and across all levels of rurality. When expanded to rare incidents such as incomplete plumbing in rural areas, it becomes even more severe. We apologize this point was not made clear in our initial draft. We have significantly expanded upon this concern in our manuscript and this addition is repeated below.

Beyond census data, the data from ECHO on water quality is difficult to work with and poorly maintained by the EPA, requiring us to be creative in answering our research questions. This means that we were forced to use a variety of procedures for geolocating community water systems and Clean Water Act permittees. As detailed in the manuscript, in some cases county was actually

listed, but in others we relied significantly upon reported latitude and longitude. Unfortunately, these point locations, particularly for community water systems, do not reflect the full reach of a water system. Thus, whatever spatial unit we choose to use carries the nontrivial assumption that the point location of a system services people within that unit. This issue is similar to that raised by Mohai and Saha (2006) about distance from a hazard. Since water is regulated at the county level (e.g. National Association of Counties, 2017), we believe this assumption makes sense if we analyze relationships at the county level. Unfortunately, since census tracts are completely apolitical, we are not comfortable making this assumption at the census tract level.

The issue you raise is important and one we spent a great deal of time considering to determine if we felt that a smaller scale could be used appropriately. While we decided not to pursue that course of action, we hope this justification is sufficient. Further, we have taken strides to incorporate every other suggestion you made. Finally, we also acknowledge this important information was absent in our first draft and have added it into our manuscript. It is repeated below:

“Similar to prior work in this area,^{4,5,8} we restrict our analysis to the scale of the county for reasons related to data limitations and resulting conceptual validity. Although counties are arguably larger in geographic area than ideal for an environmental injustice analysis, if we were to use a smaller unit for which data is available such as the census tract, the conceptual validity of the analysis would be limited due to the apolitical nature of these units. As outlined above, ECHO data is messy and missing many geographic identifiers. What is provided is generally either the county or latitude and longitude. If only the county is provided, then we are constrained to using the county regardless of conceptual validity. However, even when latitude and longitude are provided—which is the case for many observations—the provided point location says nothing about which households the water system or permittee is serves or impacts. Due to this, whatever geographic unit we use carries the assumption that those in the unit could be plausibly impacted by the water system or permittee. Given that counties are often

	responsible for both regulating drinking water, as well as maintaining and providing water infrastructure,²⁹ we were comfortable with this assumption between point location and presumed spatial impact when using the scale of the county. However, we believe this assumption would have been invalid and untestable for smaller apolitical units for which demographic data is available such as census tracts. Beyond the issues presented by ECHO data, the county is also the appropriate scale of analysis for this study due to the estimate-based nature of the ACS. ACS estimates are based on a rolling five-year sample structure and often have very large margins of error. At the census tract level, these standard errors can be massive, especially in rural areas.³⁰⁻³² Due to this variation, and the need to include all rural areas in this analysis, the county, where the margins of error are considerably smaller, is the appropriate unit for this study. All of this said, the county is, in fact, a larger unit than often desired or used in environmental justice studies. Studies focused on exclusively urban areas with clearer pathways of impact can and should use smaller units such as census tracts. It will be imperative for future scholarship focused on water hardship across the rural-urban continuum to gain access to reliable data on sub-county political units, as well as data linking water systems to users, to continue documenting and pushing for water justice.”
3-Although multivariate regression models are presented and used to draw important conclusions in this study, there is no discussion or examination of multicollinearity. Since most of independent variables in the models presented in Table 2, 3, and 4 are highly likely to be strongly correlated with each other (e.g., median household income vs. poverty rate) at the county level, it is important to analyze this issue and indicate how it was addressed for the multivariate analysis. 4-More justification is necessary on the explanatory variables chosen to measure the distributive EJ implications. The authors should consider using variables that comprise the widely used and CDC-recommended social vulnerability index (Flanagan et al. 2018: https://www.ncbi.nlm.nih.gov/pmc/articles/PMC7179070/), instead of using an arbitrary subset of these variables. It is important to at least consider	Thank you for this point. You are correct we neglected to fully assess this issue in our first draft. We have now assessed multicollinearity in our models. As a result, we have removed percent non-Latino/a white as a variable. All other variables, including median income and poverty, did not display problematic levels of multicollinearity (e.g. VIF > 10). We now clarify this within our methods. Thank you for this. The selected variables were not arbitrary, but we admit they were poorly articulated in the initial draft. We have added additional justification of these variables and now relate each variable to the relevant dimensions of the social vulnerability index. The variables we use are also outlined in the introduction as previously documented dimensions of

WATER HARDSHIP

how the selected variables represent the domains established there.	environmental injustice. Some of this language is repeated below: “To assess this social clustering, we estimate linear probability models of elevated levels of incomplete plumbing and poor water quality with the previously identified environmental justice dimensions of age, income, poverty, race, ethnicity, education, and rurality as our independent variables. We include these independent variables due to their prevalence within prior work on environmental injustice in both rural and urban areas.¹⁷⁻²⁵ Further, although there is not a one-to-one overlap, these variables conceptually map onto the dimensions of the Center for Disease Control Social Vulnerability Index: Socioeconomic Status (i.e. income, poverty, education), Household Composition & Disability (i.e. age), Minority Status & Language (i.e. race and ethnicity), and Housing & Transportation (i.e. rurality).²⁸”
5- On a related note, I found some of the choices for explanatory variables (e.g., % Black and % White) to be problematic. Since the White and Black categories from the U.S. Census or ACS include both Hispanics and non-Hispanics, only % non-Hispanic White and % non-Hispanic Black should have been used to avoid double counting problems. This would also make the statistical results consistent and comparable with most recent national scale distributive EJ studies conducted in the U.S.	Thank you for this point. We have revised our measures of race and ethnicity to use the mutually exclusive groupings instead of the overlapping groupings we initially used.

Reviewer 2	
Comments	Note
This is a fantastic and very much needed overview of the household water crisis in US. It is clearly written and very accessible. The most noteworthy results stem from the spatial patterning of different forms of water hardship and how these differences are unlikely to be resolved with a “one size fits all” approach. The manuscript is well situated within the literature to make significant contributions around water security, environmental justice, sustainability planning, and related fields. The methods appear robust, are clearly outlined and replicable, and I cannot discern any major flaws within the analysis.	Thank you so much for your favorable reviews. We have addressed your concerns and hope you find the paper suitable for publication.
I only have a couple of minor quibbles with the manuscript, which center around the need to make	Thank you, we have clarified what we meant in this section and point directly to studies which

WATER HARDSHIP

a few points more clear. The first is on page 2, line 72, when the authors state that “much of the work has become dated and it generally focuses on a single dimension of the issue at a time”, can they provide an example? Sure, some work is dated, but what are examples of the single dimension problem?	only assessed one dimension of the problem at a time. This new language is repeated below: “Although water hardship in the United States has experienced some academic attention, much of the work has become dated and has generally focused on a single dimension of the issue at a time—for example, recent scholarship has focused on exclusively incomplete plumbing,^{3,4,9} water quality,^{5,10} or on only urban parts of the country.²”
Second, it seems like Indigenous/non-indigenous space matters in how the household water crisis takes spatial form. A more critical and/or explicit take could be made here about longstanding injustices of settler colonialism, etc. A good place to do this might be around page 3, line 82 or page 5, line 183. I think only a sentence or two somewhere would suffice.	Thank you for this point, we have included language highlighting the legacy of settler colonialism in both instances you have suggested.
Finally, and I am not sure if there is space within the manuscript to do this, but Figure 3 is quite glaring in terms of state differences. I see the authors note that it likely speaks to variable monitoring by state, but that can’t be all that explains that, can it? It’s incredible how much some of the states (WA, WY, WV, IA, MO, WI) stand out. They’re all quite different, so it really begs the question for me around what is shared among these states. But it also raises the question of the meaningfulness of this measure, if it’s only monitoring leading to high rates of noncompliance. Would there be higher rates of noncompliance everywhere with higher rates of monitoring? Anyways, I think some of these type of questions could be cleared up somewhat in the MS or added to the methods.	This is a good point and we appreciate you raising it. Although this is not visible in our mapping of the issue, the use of state fixed effects in our models controls for this issue of state-level variation. We have tried to make this section of the methods clearer. While we do not conduct further analysis to try and discern what is going on with these states, we now briefly return to it in the discussion as an important direction for future work.
Generally, I think this is a great manuscript and I can’t wait to use it for teaching and research purposes. It will be a great resource. I see it nearly publishable as is and am very thankful for the opportunity to review it.	Thank you!

Reviewer 3	
Comments	Note
This paper investigates the prevalence of access to clean drinking water in the United States. The authors use EPA reports of water quality violations and ACS estimates of complete plumbing as measures of water quality and plumbing respectively. They subsequently perform a statistical analysis to determine any socio-	Thank you for your thorough and helpful review. We have addressed your comments and concerns and hope you now find the paper suitable for publication.

WATER HARDSHIP

demographic factors which associate with clean water access issues. Overall, the paper is well written, clear, and methodologically sound. I have a few larger comments in the interest of strengthening the manuscript and minor details.	
Major comments (in the order they arise when reading): My first major comment is about the framing of the study. From the summary and the introductory body of text, the results pertaining to the scope of the lack of complete plumbing are sold as the novel aspect of the study (Lines 37-40 and 53-57 for example). This is just reporting the data provided by the ACS and—in the opinion of this reviewer— does not represent the contribution of the paper. I ask that either appropriate language be included to make it clear this data already exists (e.g. “Data from the American Community Survey shows XYZ houses are without access to complete plumbing...”) or more favorably, the introductory sections be framed to highlight other aspects of the work such as the factors associated with clean water violations or complete plumbing access.	Thank you for this point, we have adopted a hybrid of your two suggestions for reworking our framing. We now clarify that the descriptive data on incomplete plumbing already exists and highlight the contribution of the study in the form of the prevalence of water quality issues and social factors associated with water hardship.
My second major comment is about 1/100 cutoff used to develop the dichotomous statistical models. I don’t disagree with this notionally, but further justification is needed to pick 1/100 as a cutoff. One route would be to show the distribution of the county-level fraction of the population with incomplete access to plumbing and show where 1% falls in that distribution From a quick look at ACS data, 2757 counties have between 0-1% of the household lacking plumbing with 462 above that threshold. Another route would be to compare the results of the statistical model at multiple thresholds.	We agree our justification for this cutoff was not sufficient in our first draft. We have added additional justification for the 1/100 threshold in the methods, which is repeated below. We ground this justification in the Sustainable Development Goals laid out by the UN. While we did investigate the distributions for our outcomes—for example just about all of the 509 counties over the plumbing thresholds were outliers in a standard box and whisker plot—we did not decide to pursue a data-driven cutoff or justification because we believe doing so would not be independent from the prevalence of the outcome. For example, if we were to pick a cutoff based upon a break in the data or some degree of standard deviations, this would be subject to whether or not the outcome is currently at an elevated rate everywhere. Thus, if 1/100 should be viewed as elevated, but the issue is above an acceptable standard everywhere, we would be estimating where it is relatively elevated, not absolutely elevated against some acceptable standard. Since Sustainable Development Goal 6.1 specifies that

WATER HARDSHIP

	we need to achieve “universal and equitable access to safe and affordable drinking water for all”, we believe that the 1/100 standard represents a level that is both deemed unacceptable under model, while also being intuitive and interpretable to readers. While we could have dipped below 1/100, we felt the nominal value of the measure made keeping it worthwhile. We hope you will find this sufficient. We now relate the cutoff to SDG 6.1 and the logic we have outlined here. Which is repeated below: “For models of water injustice, a dichotomous measure which classified counties as either having low levels of the specific water issue or elevated levels or the specific water issue, was used due to the low relative frequency of water access and quality issues relative to the whole United States population. For all three outcomes, we benchmark an ‘elevated’ level of the issue as what would be viewed as an unacceptable level under United Nations Sustainable Development Goal 6.1, which states, “By 2030 achieve universal and equitable access to safe and affordable drinking water for all”.¹ As this goal focuses on ensuring all people have safe water, we deem a county as having an elevated level of the issue if greater than one percent of households, community water systems, or permittees had incomplete plumbing, were in Significant Violation, or Significant Noncompliance, respectively. Although we could have used an even stricter threshold given the SDG’s emphasis on ensuring access for all people, we use one percent as our cutoff due to its nominal value and ease of interpretation.”
My third major comment is about the use of population-level statistics. In lines 106-107 and 123-126, the statistics presented are the total population residing in counties in which over 1% of the population is in water crisis. This is borderline-misleading the way it is presented, as it would be easy to mistake this for the total population that did not have indoor plumbing. This is more an issue when used in lines 123-126, as the results moves from percentages directly to the ‘fraction in the county’ numbers. I understand the utility of this measurement as a way to describe the prevalence of this issue geographically (i.e. saying this is something that could be occurring in your community). However,	We apologize if the results appeared misleading. We have revised this section to be precise in what is being presented and what each statistic represents. We still include this statistic within the results because it is not possible to give the exact number of people impacted by a CWA Noncomplier due to information linking number of users to utilities being unavailable. That said, we fully appreciate your concerns with how we initially presented this statistic and have drafted much more transparent language. This new language is repeated below: “Further, 509 counties, representing over 13 million Americans, have an elevated level of the issue where greater than one percent of household

WATER HARDSHIP

further clarification is needed, preferably by stating clearly a percentage of the population impacted followed by the absolute number impacted and moving this type of broader impact statistic to the discussion.	do not have complete indoor plumbing. Thus, even if individuals are not experiencing the issue themselves, they may live in a community where incomplete plumbing is a serious issue.” “Due to limitations in the data, we are unable to determine exactly how many individuals are linked to each problematic community water system or Clean Water Act permittee, however, we do find that over 81 million Americans live in counties where more than one percent of community water systems are listed as Significant Violators, and 217 million Americans living in counties where greater than one percent of Clean Water Act permittees are Significant Noncompliers. Thus, although the number of individuals impacted by these issues is certainly far smaller than these totals, a vast number of Americans live in communities where issues of water quality are elevated.”
Minor comments: With American Indian and Alaskan Native populations found to be more likely to have incomplete plumbing and incomplete water, there may be more at play as a result of federal policies which place significant limitations on what development can be done on native lands and may contribute to lack of access to plumbing.	This is a valuable point and we have added it as a possible explanation.
Line 118: is it possible to determine the overlap between CWA noncompliant systems and SDWA violators?	Unfortunately no, the data does not facilitate this kind of comparison.
Lines 331-332: Methodologically, I fully understand why dropping data without reported lat/lon is a reasonable decision. I worry, however, that there may be association between missing data and noncompliant facilities. Poor management or lack of institutional knowledge of EPA regulations and compliance immediately come up as reasons why a facility may be both non compliant and reporting incomplete data. If feasible, this reviewer would like to see a brief summary of the rate of compliance of facilities which were dropped (both for SDWA and CWA). If they are atypically high, this furthers the case of the paper that this is an	We have done this supplemental analysis and now report the results within our methods. For the SDWA data, there did not appear to be atypically high or low. For CWA data there were only 10 cases dropped, which while having a high level of noncompliance (7/10), it did not appear to try and glean anything from this small of an N.

WATER HARDSHIP

under-reported issue. If they low, this should be included as a discussion point.	
Figures 1,2 and 3 are difficult to assess trends from in areas dense with counties. I suggest the following changes (1) use a lighter color for the county borders so areas like northeastern TN, western NC, central KY and southern IN are more easily identified. This should also help the visual congestion in coastal areas. (2) Use a consistent scale throughout. I'm unsure if it's an issue with a map projection or image size, but currently Puerto Rico looks to be about the size of Mississippi which is certainly not true in reality (3) Eastern Maine is cut off	Thank you for these suggestions. We have incorporated them to improve our figures. We shrunk Puerto Rico, but did not put it to scale because if we put AK, PR, HI, and the rest of the USA all in one scale, the map would be either uninterpretable or massive.

Reviewer comments, second round

Reviewer #1 (Remarks to the Author):

The authors have adequately addressed my key concerns with the previous of this manuscript and made appropriate revisions to justify important choices regarding variables and methodology. My only remaining concern focuses on how they have assessed multicollinearity in their models. Lines 341-342: "Further, we assessed all full models for multicollinearity via variance inflation factor (VIF) and all variables in all models had acceptable VIF values below 10.0." VIFs are not very effective in fully capturing the severity of multicollinearity in a multivariable model since it focuses on individual variables. Why not report the condition index instead? This will be based on the collection of all independent variables included in the model and will truly indicate whether or not there are serious problems of collinearity that warrant some attention in the analytical process. This multicollinearity condition index for each model should be reported in the text and Tables S2, S3, and S4.

Reviewer #2 (Remarks to the Author):

Thanks again for this manuscript. I look forward to using it in the future. All of my comments have been adequately addressed.

Reviewer #3 (Remarks to the Author):

I commend the authors on the thoroughness of their review responses both to my and the other reviewers' concerns; particularly with regards to concerns of county vs tract level of analysis.

I have no further comments, and appreciate the authors's time.

WATER HARDSHIP

Reviewer 1	
Comments	Note
The authors have adequately addressed my key concerns with the previous of this manuscript and made appropriate revisions to justify important choices regarding variables and methodology. My only remaining concern focuses on how they have assessed multicollinearity in their models. Lines 341-342: “Further, we assessed all full models for multicollinearity via variance inflation factor (VIF) and all variables in all models had acceptable VIF values below 10.0.” VIFs are not very effective in fully capturing the severity of multicollinearity in a multivariable model since it focuses on individual variables. Why not report the condition index instead? This will be based on the collection of all independent variables included in the model and will truly indicate whether or not there are serious problems of collinearity that warrant some attention in the analytical process. This multicollinearity condition index for each model should be reported in the text and Tables S2, S3, and S4.	Thank you for your note. We have done as requested and now report the condition index as well. We include both because the condition index cannot be calculated for categorical variables, meaning we needed VIF values for our rural indicator. The condition index for our models—which is the same across all three multivariable models because the condition index only factors in the independent variables which are the same in each model—was 5.4, far below the conservative cut-off of 15 and dramatically below the conventional cut off of 30. Thus, we deem multicollinearity as not an issue in these models. As requested, we now report the value in the text and alongside each table. The new text is as follows: “Further, we assessed all full models for multicollinearity via condition index and VIF values and the independent variables had an acceptable condition index of 5.48, well below the conservative cut-off of 15, as well as VIF values of less than 10.”
Reviewer 2	
Comments	Note
Thanks again for this manuscript. I look forward to using it in the future. All of my comments have been adequately addressed.	Thank you!
Reviewer 3	
Comments	Note
I commend the authors on the thoroughness of their review responses both to my and the other reviewers' concerns; particularly with regards to concerns of county vs tract level of analysis. I have no further comments, and appreciate the authors's time.	Thank you!